**Investigation**

# Copy number variations and their effect on the plasma proteome

Daniel Schmitz,[1],* Zhiwei Li,[1] Valeria Lo Faro,[1] Mathias Rask-Andersen,[1] Adam Ameur,[1] Nima Rafati,[2] Åsa Johansson[1],*

[1]Department of Immunology, Genetics and Pathology, Science for Life Laboratory, Uppsala University, Box 815, 751 08 Uppsala, Sweden
[2]Department of Medical Biochemistry and Microbiology, National Bioinformatics Infrastructure Sweden, Science for Life Laboratory, Uppsala University, Box 582, 751 23 Uppsala, Sweden

*Corresponding author: Department of Immunology, Genetics and Pathology, Science for Life Laboratory, Uppsala University, Box 815, 751 08 Uppsala, Sweden. Email: Daniel.schmitz@igp.uu.se; *Corresponding author: Department of Immunology, Genetics and Pathology, Science for Life Laboratory, Uppsala University, Box 815, 751 08 Uppsala, Sweden. Email: Asa.johansson@igp.uu.se

Structural variations, including copy number variations (CNVs), affect around 20 million bases in the human genome and are common causes of rare conditions. CNVs are rarely investigated in complex disease research because most CNVs are not targeted on the genotyping arrays or the reference panels for genetic imputation. In this study, we characterize CNVs in a Swedish cohort ($N = 1,021$) using short-read whole-genome sequencing (WGS) and use long-read WGS for validation in a subcohort ($N = 15$), and explore their effect on 438 plasma proteins. We detected 184,182 polymorphic CNVs and identified 15 CNVs to be associated with 16 proteins ($P < 8.22 \times 10^{-10}$). Of these, 5 CNVs could be perfectly validated using long-read sequencing, including a CNV which was associated with measurements of the osteoclast-associated immunoglobulin-like receptor (OSCAR) and located upstream of *OSCAR*, a gene important for bone health. Two other CNVs were identified to be clusters of many short repetitive elements and another represented a complex rearrangement including an inversion. Our findings provide insights into the structure of common CNVs and their effects on the plasma proteome, and highlights the importance of investigating common CNVs, also in relation to complex diseases.

Keywords: copy number variation; plasma proteome; whole-genome sequencing; long-read sequencing

## Introduction

Genome-wide association studies (GWAS) have enabled the discovery of thousands of associations between single-nucleotide polymorphisms (SNPs) and human traits as well as diseases. Despite their success, GWAS can only explain a fraction of observed heritability. One study estimated the variance of 32 complex traits explained by common SNPs to lie between 9.8 and 48.9% (Nolte et al. 2017). However, the heritability behind these traits is higher, suggesting that additional genetic effects not captured by common SNPs exist. These effects might be explained by structural variants (SVs), which are genomic alterations affecting at least 50 bp. In 2015, the 1,000 Genomes project consortium found that a typical genome contains 2,100 to 2,500 SVs covering 20 million bases (Auton et al. 2015). Copy number variations (CNVs) are SVs that affect the number of copies of a given genomic region an individual carries. CNVs can cause dosage imbalance of the genes involved, bias GWAS effect estimates of proximal SNPs and disrupt linkage disequilibrium (LD) patterns (Handsaker et al. 2015; Khrunin et al. 2016; Liu et al. 2018). Only 73% of all common CNVs are in linkage disequilibrium (LD) with a nearby SNP (Sudmant et al. 2015), and therefore, CNVs are likely to capture additional genetic effects on top of the SNPs identified in a GWAS. This emphasizes the importance of analysing CNVs in relation to human traits and diseases to uncover the missing heritability that cannot be captured in the large-scale GWAS that has been performed.

Few association studies have considered the effect of SVs on common diseases and complex traits. Previously, CNVs have been associated with glaucoma (Lo Faro et al. 2021), psychiatric disorders (Huang et al. 2012; Malhotra and Sebat 2012; Martin et al. 2020), Crohn's diseases, type 1 diabetes, and multiple developmental diseases (Huang et al. 2012; Coe et al. 2014; Liu et al. 2020). Several studies linked both germline and somatic CNVs to multiple types of cancer, including an integrated analysis of CNVs, SNPs, and expression data (Park et al. 2015; Momtaz et al. 2018; Brezina et al. 2020; Chattopadhyay et al. 2021). A previous study using data from UK Biobank found CNVs in 28 genes to be associated with 13 blood biomarkers (Sinnott-Armstrong et al. 2021). A different study in a cohort of 1,457 individuals has found 4 genome-wide associations between CNVs and protein levels (Png et al. 2020).

Traditionally, array-based methods, such as SNP arrays and array comparative genomic hybridization, have been used to detect genetic variations, including CNVs. However, they are limited to previously selected regions and usually cannot detect CNVs shorter than 8 kbp (Quenez et al. 2020). Quantitative polymerase chain reaction (qPCR) and digital droplet polymerase chain reaction (ddPCR) have been established as high-yield, low-cost alternative protocols for CNV calling but require the design of customized assays and cannot resolve complex structural variation (Weaver et

al. 2010; Mazaika and Homsy 2014). Nowadays, sequencing technologies have enabled accurate detection of genetic variation not limited to SNPs that are available on a genotyping array or in an imputation reference panel (Höglund et al. 2019; Gilly et al. 2020). With improvements in data quality and cost, whole-genome sequencing (WGS) has become more popular in large-scale genomic studies. In addition to identifying SNPs, these methods also have the potential to identify SV. Especially long-read sequencing, such as Pacific Bioscience (PacBio) Single-Molecule Real Time (SMRT) sequencing, which is seeing growing adoption, promises to accurately detect SVs and CNVs.

Plasma proteins are well-studied traits in GWAS, as they represent the intermediate layer between genetic variation and disease development. Several studies have identified protein quantitative trait loci (pQTLs) in population-based cohorts and linked their findings to downstream disease risk (Enroth et al. 2014; Enroth et al. 2015; Sun et al. 2018; Sinnott-Armstrong et al. 2021).

In this project, we focused on characterizing CNVs from high coverage WGS data of over 1,000 individuals from the Northern Sweden Population Health Study (NSPHS) (Höglund et al. 2019). We called CNVs using CNVnator, and tested for association between CNVs and the variation in a large set of proteins ($N = 438$) that represent well-established or exploratory biomarkers of disease. Subsequently, we resequenced 15 individuals from our cohort using SMRT technology, to verify the CNVs at individual-level.

## Materials and methods
### Study cohort
The Northern Swedish Population Health Study (NSPHS) is a cohort study of 1,047 individuals conducted in 2 municipalities in the region of Norrbotten, Sweden. All participants gave their written informed consent. In cases where the participant was not of age, a legal guardian signed additionally. The study was performed in compliance with the Declaration of Helsinki. The WGS was performed at SciLifeLab in Stockholm on an Illumina HiSeq (X-Ten) platform at 30× coverage, and mapped to the reference genome GRCh37, as described previously (Ameur et al. 2017; Höglund et al. 2019). After quality control (QC), which was described in previous publications (Höglund et al. 2019), a total of 1,021 samples remained for analysis. Proteins were measured in 903 individuals of the NSPHS cohort using the Olink Protein Extension Assay (PEA) and 5 Proseek panels (CVD2, CVD3, NEU1, ONC2, INF1), as described previously (Enroth et al. 2018). In short, PEA is an affinity-based assay that uses oligonucleotide-labelled pairs of antibodies that bind to the target proteins in close proximity to each other. If both antibodies bind, they produce a polymerase chain reaction (PCR) target sequence, which can be quantified using standard real-time PCR. The analysis was performed on plates with 96 wells, allowing for 92 individuals as well as 3 positive and one negative controls per batch, which serve to determine the lower detection limit and normalize the protein measurements. Signals below the detection limit were removed. The complete QC procedure was described previously (Enroth et al. 2014; Höglund et al. 2019). The remaining measurements were adjusted for age, sex, and batch effects and normalized using the rank-based inverse normal transformation (mean = 0, standard deviation = 1). A total of 438 proteins and 872 samples passed both genotyping and protein QC (Fig. 1).

### CNV calling using CNVnator
For our primary analysis, we used WGS data generated on an Illumina HiSeq (X-Ten) platform at 30× coverage. We called CNVs using CNVnator version 0.4.1, which employs a binning approach and reports copy numbers (CNs) as continuous values corresponding to the factor by which the read depth in the CNV differs from its surrounding regions (Abyzov et al. 2011). We first used CNVnator to estimate the optimal bin size in bp for each sample, as the lowest value of from the set (70, 85, 100, 150, 200, 250) at which the ratio of the bins' read depth (RD) mean value to the standard deviation was between 4 and 5. The reason for having the mean value of the converted RD signal 4 to 5 times greater than its standard deviation is to preserve enough statistical power for detecting deletions by $t$-test between the regional and global RD signal while detecting the variations with the smallest bin size possible to enable breakpoint detection at a higher resolution (Abyzov et al. 2011). CNVs were then identified in each sample separately and filtered in accordance with the recommendations by the CNVnator's authors (Abyzov et al. 2011). CNVnator provides $P$-values for each detected CNV calculated from a 1-sample $t$-test between the local and global RD signal as well as a real-valued CN estimate quantifying the factor by which the CNV's RD differs from its surrounding regions. In the initial CNV calling, CNVs were considered high-quality detections if they passed the significance threshold $P < 0.05$ after Bonferroni correction for number of samples and the number of CNVs identified in each respective sample. Additionally, we excluded CNVs where the fraction of reads with low quality and ambiguous alignments ($q_0$) within a call region was 0.5 or higher. After the CNV discovery, we used BEDTools version 2.29.2 to create a CNV matrix (Quinlan and Hall 2010). First, the genome was split into non-overlapping 200 bp windows, and all windows, for which any sample had a high quality CNV detected, were included in the CNV matrix. To determine each individual's CN of these selected CNVs, we applied less stringent QC for inclusion, with only the $q_0$ threshold applied. Samples for which no CNV had been identified within a 200-bp window were assigned CN 2 (wild-type) and samples that failed QC were set to "NA" for that window. Finally, adjacent 200-bp windows with identical CNs across all samples were merged. The merged windows represented the final set of CNVs used in our downstream association analyses.

### Association analysis
In the association analysis, we tested for association between the CN at each of the 184,182 CNVs and the measurements of the 438 plasma proteins. We used linear regression models in the glm function in R (version 4.3.4) with the CNs (as a continuous value as reported by CNVnator) for the CNV as a predictor and the protein measurement as response. We included the first 4 genetic principal components (PCs) as covariates in the model. The PCs were calculated from SNP genotypes (Kierczak et al. 2022). We have previously estimated the power to detect associations for rare variants at a genome-wide significance threshold in the NSPHS cohort (Höglund et al. 2019). Based on that, we excluded all combinations of CNVs and proteins from our association testing, for which less than 3 individuals had both a CN different from 2 and a protein measurement available. We applied a Bonferroni adjustment for multiple testing and adjusted for the number of tests we performed after filtering (60,814,115), resulting in a significance threshold of $P < 8.22 \times 10^{-10}$.

After the association analysis, we identified independent signals by clumping nearby CNVs that passed the significance threshold and whose CNs were highly correlated ($R^2 > 0.8$). CNVs on the same chromosome with $R^2 < 0.1$ were considered independent. The resulting clumped CNVs formed the basis for the reported results.

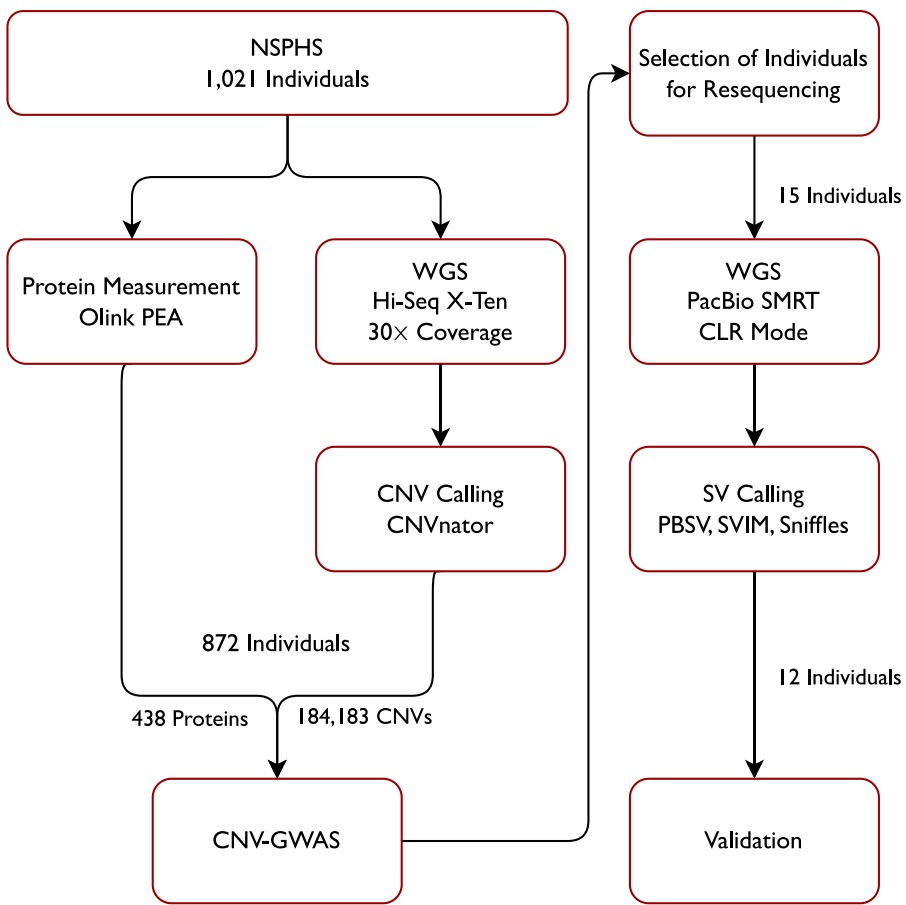

**Fig. 1.** Flow chart of the analysis pipeline. Whole-genome sequencing (WGS) of 1021 individuals was performed of which 872 passed the quality control (QC) for the copy number variant (CNV) calling and for the measurements of 438 proteins. A subset of the individuals was selected for resequencing using the long-read Pacific Bioscience (PacBio) Single-Molecule Real Time (SMRT) technology, to validate and characterize CNV regions that were significantly associated with the plasma proteome.

For all significant CNV-protein associations, we compared overlapping results with a previous GWAS in the same cohort (Kierczak et al. 2022). A potential overlap was considered if the same protein was associated with both a CNV (in the current study; Table 1) and with an SNP in our previous GWAS ($P < 5 \times 10^{-8}$), and the SNP and CNV were located within 1 Mbp of each other. The $P$ value to select potential overlapping associations from the GWAS was less conservative in order to not miss any potentially overlapping signal. For all overlapping associations, the LD between the lead SNP from the GWAS as well as conditional SNPs passing the significance threshold were estimated by calculating the Spearman rank correlation between allele counts and CNs. Furthermore, we performed conditional analyses by adjusting for SNPs with overlapping associations.

Additionally, we performed a sensitivity analysis, which included 10 instead of 4 genetic PCs, and investigated the significant CNVs for overlaps with known repeats using the RepeatMasker track from UCSC Genome Browser (Kent et al. 2002; Nassar et al. 2023). We estimated genomic inflation factors for all proteins independently and one joint inflation factor. We tested whether the genomic inflation factors of proteins with significant associations were higher than those without using a Kruskal–Wallis rank sum test.

### Resequencing with PacBio SMRT

We selected 15 individuals for long-read sequencing using PacBio SMRT technology based on the results of our association analysis. Samples were selected so that as many as possible of the 15

significant CNVs had a non-reference call in any of the individuals, disregarding regions known to be problematic such as the *HLA* locus on chromosome 6. We performed WGS on a PacBio SEQUEL II system in continuous long read (CLR) mode. Libraries were prepared according to manufacturer specifications. The resulting reads underwent standard QC procedures and were mapped to human genome assembly GRCh37 (the same build as the Illumina data) using pbmm2 version 1.4.0. We then called structural variation using SVIM v1.4.2, Sniffles v1.0.12, and pbsv v2.4.0 and visualized the coverage and reads supporting the CNVs in the regions of interest using samplot v1.1.0 (Belyeu et al. 2021).

We excluded all samples whose unique molecular yield was less than 30 Gbp, corresponding to $10 \times$ coverage, which resulted in 12 individuals remaining. We called structural variation using 3 different tools: Sniffles, SVIM, and pbsv. We considered calls with 50% reciprocal overlap and alleles that implied a consistent CN change with CNVnator calls to be equivalent. For a secondary analysis, we only required consistent calls to overlap by 1 bp. We proceeded by manually checking the CNVs with significant associations for evidence of their existence.

## Results
### CNV detection

The mean optimal bin size for CNV calling with CNVnator was 91 bp (range: 70–150, median: 85). The size of the CNVs reported

**Table 1** Significant CNV-protein associations after adjusting for the number of CNVs identified.

| CNV # | Coordinates (GRCh37)[a] | Width (bp) | Protein | Coding gene coordinates[b] | Lead-CNV[c] | Beta | SE | P | CN range (Min–Max) | % carriers (deletions/ duplications)[d] |
|---|---|---|---|---|---|---|---|---|---|---|
| 1 | 1:159,018,151–159,018,951 | 800 | WFIKKN1 | 16:679,238–684,116 | 1:159,018,751–159,018,951 | 0.85 | 0.13 | 1.61E-10 | 0.08–2.63 | 6/0 |
| 2 | 1:179,455,601–179,455,801 | 200 | LY9 | 1:160,765,863–160,798,051 | 1:179,455,601–179,455,801 | 0.55 | 0.09 | 6.15E-10 | 0.00–2.00 | 15/0 |
| 3[e] | 3:98,410,601–98,414,801 | 4,200 | ICAM-2 | 17:62,079,953–62,097,994 | 3:98,410,601–98,410,801 | 0.62 | 0.04 | 4.17E-45 | 0.00–2.00 | 63/0 |
| 3[e] | 3:98,410,601–98,414,801 | 4,200 | CD200R1 | 3:112,640,055–112,693,969 | 3:98,411,801–98,413,201 | 0.46 | 0.05 | 2.52E-21 | 0.00–2.00 | 68/0 |
| 3[e] | 3:98,410,601–98,414,801 | 4,200 | PD-L2 | 9:5,510,569–5,571,254 | 3:98,411,201–98,411,401 | 0.39 | 0.05 | 6.82E-16 | 0.00–2.00 | 68/0 |
| 3[e] | 3:98,410,601–98,414,801 | 4,200 | Siglec-9 | 19:51,628,164–51,639,908 | 3:98,411,201–98,413,201 | 0.67 | 0.04 | 4.71E-46 | 0.00–2.00 | 68/0 |
| 3[e] | 3:98,410,601–98,414,801 | 4,200 | VEGFR-3 | 5:180,028,505–180,076,624 | 3:98,413,601–98,413,801 | 0.59 | 0.04 | 2.03E-36 | 0.00–2.00 | 67/0 |
| 4 | 5:70,391,301–70,394,701 | 3,400 | IL-18 | 11:112,013,973–112,034,840 | 5:70,393,101–70,393,301 | 0.37 | 0.06 | 5.41E-11 | 0.79–5.06 | 1/12 |
| 5 | 6:31,193,201–31,194,401 | 1,200 | MIC-AB | **6:31,367,560–31,383,092** | 6:31,193,201–31,193,401 | 0.65 | 0.09 | 6.80E-13 | 0.00–2.00 | 9/0 |
| 6 | 6:31,337,891–31,341,891 | 4,000 | MIC-AB | **6:31,367,560–31,383,092** | 6:31,338,291–31,338,891 | −1.04 | 0.11 | 1.12E-18 | 0.00–2.00 | 9/0 |
| 7 | 6:32,450,801–32,453,801 | 3,000 | MIC-AB | **6:31,367,560–31,383,092** | 6:32,450,801–32,451,001 | 0.39 | 0.05 | 2.71E-13 | 2.00–5.04 | 0/49 |
| 8 | 6:32,496,001–32,497,401 | 1,400 | MIC-AB | **6:31,367,560–31,383,092** | 6:32,496,601–32,496,801 | 0.59 | 0.07 | 4.33E-17 | 0.13–2.00 | 33/0 |
| 9[f] | 6:32,522,201–32,523,601 | 1,400 | CCL19 | 9:34,689,563–34,691,274 | 6:32,523,401–32,523,601 | −0.47 | 0.05 | 2.20E-17 | 0.07–2.00 | 33/0 |
| 10 | 11:67,330,156–67,332,356 | 2,200 | FR-gamma | 11:71,825,914–71,850,936 | 11:67,330,356–67,332,156 | −1.23 | 0.11 | 9.71E-26 | 0.02–2.00 | 8/0 |
| 11 | 16:28,611,246–28,624,046 | 12,800 | ST1A1 | **16:28,616,516–28,634,946** | 16:28,613,646–28,613,846 | 0.77 | 0.07 | 2.67E-25 | 0.92–5.17 | 1/22 |
| 12 | 17:36,387,671–36,399,671 | 12,000 | CCL4 | **17:34,430,982–34,433,014** | 17:36,392,671–36,394,671 | 0.15 | 0.02 | 7.86E-11 | 2.00–13.36 | 0/92 |
| 13 | 17:39,203,601–39,211,001 | 7,400 | CCL15 | 17:34,323,475–34,329,084 | 17:39,203,801–39,209,001 | −1.04 | 0.13 | 5.90E-16 | 0.06–2.00 | 6/0 |
| 14 | 19:41,381,791–41,387,591 | 5,800 | MIA | **19:41,277,552–41,283,395** | 19:41,381,991–41,385,191 | −1.23 | 0.17 | 2.91E-12 | 0.33–3.05 | 4/0 |
| 15 | 19:54,555,501–54,560,501 | 5,000 | hOSCAR | **19:54,597,932–54,606,000** | 19:54,558,901–54,559,101 | −0.71 | 0.06 | 5.04E-27 | 0.00–2.00 | 31/0 |

a Largest region where all windows reached genome-wide significance and were in LD ($R^2 > 0.8$) with the lead CNV, i.e. the CNV with the strongest association.
b Associations in cis (distance < 2 Mbp) are displayed in bold.
c CNV with the strongest association where all 200-bp windows were assigned consistent CNs across the population. These were used in the association analysis.
d Fraction of individuals called with a deletion (CN < 2) and duplication (CN > 2) in the whole cohort.
e In LD with SNPs rs1093473 and rs11927405, which have the same association.
f In LD with SNP 6:32668773 with the same association.
The table shows the summary statistics of the most significant 200-bp window in each CNV. Abbreviations: CNV, copy number variations; SE, standard error; CN, copy number; P, P-value.

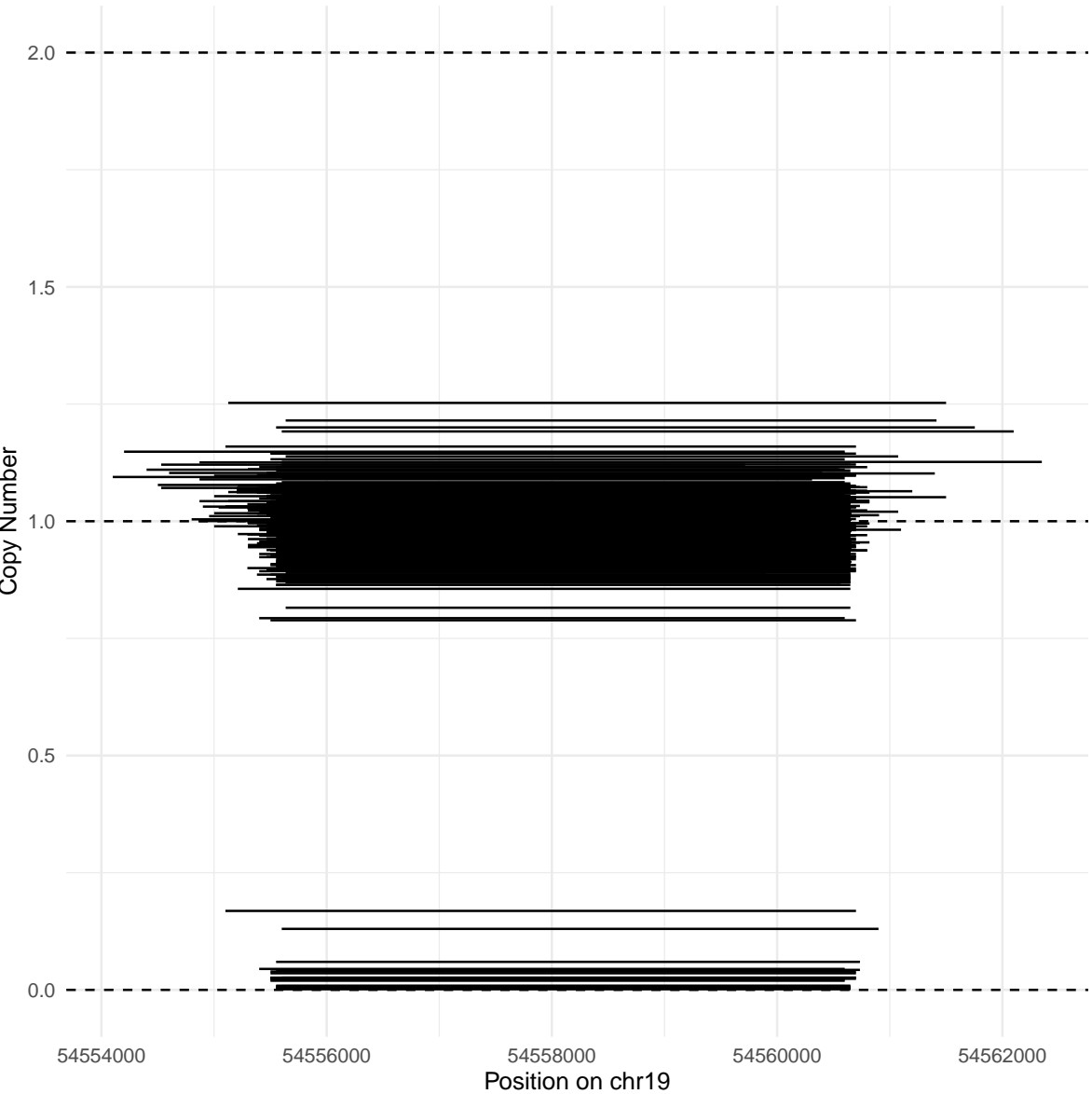

**Fig. 2.** Example of CNVs reported by CNVnator. The deletions cover up to 8 kbp on Chr19. Each line represents the CN for 1 sample in the cohort as estimated from read depth by CNVnator. Only samples, for which a CNV was detected, i.e., that deviates from the wild-type (CN = 2, a normal diploid genome) are illustrated. The dashed lines indicate CN = 1 (heterozygous deletion). Individuals with CN = 0 indicate individuals who are homozygous for a deletion in the region. These calls were all considered to constitute the same signal in the association analysis.

per individual sample ranged from 1000 bp to 21.1 Mbp. As much as 68% of all CNVs were removed in the QC procedure, resulting in an average of high-quality deletions (CN < 2) and 638 high-quality duplications (CN > 2) per sample. The number of per-individual high-confidence CNVs we identified was lower than in previous studies but the number of duplications was comparable (Abel et al. 2020). The breakpoints of the CNVs varied between individuals (Fig. 2). The final CNV matrix contained genotypes (CN value) of 872 individuals in 2,182,133 windows of 200 bp. After merging adjacent windows with consistent CNs, considering deletions and duplications jointly, a total of 184,182 CNVs remained for downstream analyses.

## Association analysis

We detected 19 significant associations between 16 proteins and 15 independent CNVs ($P < 8.22 \times 10^{-10}$). These 15 independent CNVs consisted of 324 of the 200-bp windows and were distributed over

8 chromosomes (Fig. 3, Table 1). Eleven associations were *trans*-associations i.e. they were associated with proteins whose coding genes were on different chromosomes or more than 2 Mbp away, accounting for 7 CNVs. This is a significantly higher proportion of associations *in trans* than among what we have identified for SNP associations in the same cohort previously (*cis*: 180, *trans*: 45, Fisher's exact test $P$ value = 0.000642) (Kierczak et al. 2022).

Our results showed signs of both light inflation and deflation (Supplementary Table 1). The proteins with significant associations did not show significantly more inflation ($\bar{\lambda} = 1.12$) than those without ($\bar{\lambda} = 1.09$) (Kruskal–Wallis $P = 0.07$, Supplementary Table 1). All CNVs we found to have a significant association overlapped with at least one known repetitive element (Supplementary Table 2). A sensitivity analysis which included 10 genetic PCs confirmed our findings (Supplementary Table 3).

The first CNV (CNV 1) on chromosome 1 was associated with WAP, kazal, immunoglobulin, kunitz, and NTR domain-containing

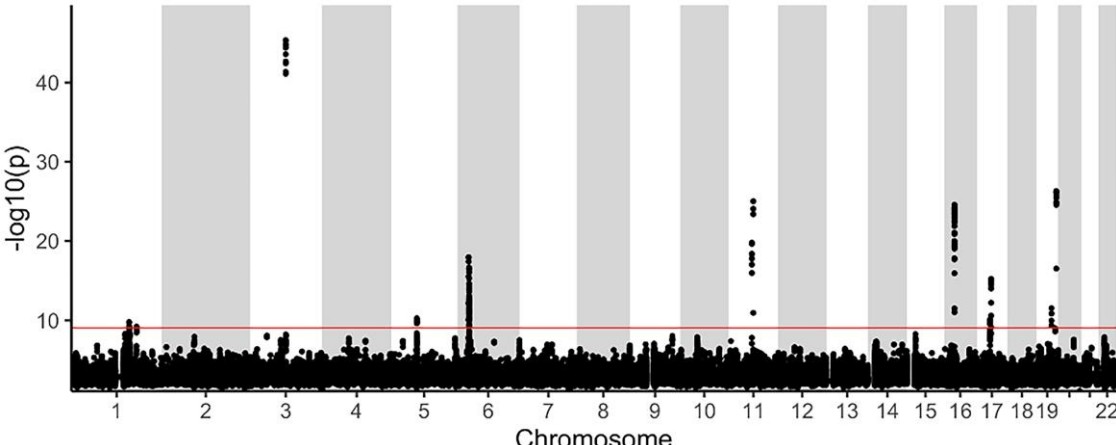

**Fig. 3.** Manhattan plot for the associations between CNVs and the protein measurements. Each dot represents a 200-bp CNV window. The *x*-axis shows the chromosomes where the CNVs are located, and the *y*-axis shows −log10 *P*-value of the association of each CNV with the 438 plasma proteins. Only the CNV-protein association with the lowest *P* value for each window is shown. The red horizontal line shows the significant threshold $P < 8.22 \times 10^{10}$. A total of 324 significant 200-bp windows, distributed over 9 loci, were associated with any of the 16 proteins.

protein 1 (WFIKKN1). CNV 2, which was located on the same chromosome but was independent ($R^2 < 0.1$) (Supplementary Table 4) of CNV 1, was associated with T-lymphocyte surface antigen Ly-9 (LY9). It was completely contained within an L1PA13 long interspersed nuclear element (LINE). On chromosome 3, there was one significant CNV, CNV 3, which was associated with 5 different proteins, the highest number of any CNV. These proteins included intercellular adhesion molecule 2 (ICAM-2), programed cell death 1 ligand 2 (PD-L2), sialic acid-binding immunoglobulin-like lectin 9 (Siglec-9), CD200 receptor 1 (CD200R1), and vascular endothelial growth factor receptor 3 (VEGFR-3). This CNV was in LD with 2 SNPs with significant associations with the same proteins: rs10935473 ($R^2 \approx 0.787$, CD200R1, PD-L2, Siglec-9, VEGFR-3), and rs11927405 ($R^2 \approx 0.788$, ICAM-2) (Kierczak et al. 2022). All associations stayed significant even after a conditional analysis which included these SNPs as covariates (Supplementary Table 5).

Chromosome 5 contained one association (CNV 4) with Interleukin-18 (IL-18). We detected 5 significant associations (CNV 5–9) on chromosome 6. All of them were located in the *HLA* locus, which is known to be a hotspot for CNVs (Saitou and Gokcumen 2020). Despite being located in close proximity to each other, none of them were in strong LD, with the highest $R^2 = 0.339$ (Supplementary Table 4). Four of these CNVs were associated with MHC class I polypeptide–related sequences A and B (MIC-AB). The remaining CNV had an association with Chemokine (C-C motif) ligand 19 (CCL19) and was in strong LD ($R^2 \approx 0.85$) with a SNP at chr6:32,668,773 which has a genome-wide significant association with CCL19 measurements (Supplementary Table 5a). The CNV retained significance even when conditioned on this SNP (Supplementary Table 5b).

Chromosomes 11 and 16 contained one association signal each. CNV 10 was associated with Folate receptor gamma (FR-gamma). CNV 11 was associated with measurements of Sulfotransferase 1A1 (ST1A1) and situated in the promotor region of *SULT1A1*, the gene that codes for ST1A1. Chromosome 17 contained 2 independent CNVs with significant associations. CNV 12 was only detected as a duplication. It had the highest maximum CN as estimated by CNVnator (13.36) and was associated with C-C motif chemokine 4 (CCL4). CNV 13 was a large deletion exclusively and associated with C-C motif chemokine 15 (CCL15).

Chromosome 19 contained 2 independent signals (Supplementary Table 4). CNV 14 was present both as a deletion and duplication.

It was associated with Melanoma-derived growth regulatory protein (MIA). Finally, CNV 15 was associated with osteoclast-associated immunoglobulin-like receptor (OSCAR).

## SMRT sequencing

The SMRT sequencing had inconsistent quality across samples with regards to the number of high-quality reads, read length, and the coverage varying substantially between samples (Supplementary Table 6). For instance, for 3 samples, less than half of all reads were high quality according to the run report. After excluding 3 individuals (pt_005_005, pt_005_006, pt_005_009) because of low coverage (<10×), 12 individuals remained for validation. In general, deletions were more consistently called among the CNV callers whereas duplications could often not be replicated. SVIM called more variations that overlapped with our CNVs than any other caller while pbsv called the fewest variations. To further characterize the CNVs, we manually reviewed all regions.

Among the 15 CNVs, 10 had been called with a CN different from 2 by CNVnator in at least 1 of the 12 individuals that passed long-read QC. Among those, several were successfully validated with SMRT sequencing (Table 2, Supplementary Table 7). Overall, we could validate 4 (CNV 3, 6, 10, and 15) of the 10 CNVs that were represented in the resequenced individuals to 100%.

CNV 3 (chr3: 98,410,601–98,414,801) was called a 4,200-bp deletion in the Illumina data of 8 selected individuals (4 heterozygous and 4 homozygous) of the subcohort. SVIM's calls agreed with CNVnator in 6 individuals while pbsv and Sniffles did not call any variants in this region. The individuals without SVIM calls had insufficient coverage in the region. Therefore, we considered CNV 3 validated by SVIM. The CNV could be validated with read-level evidence in 6 of the 8 individuals where SVIM called deletions. The remaining 2 individuals had low coverage in this area, which did not allow for SV calling. This CNV mapped perfectly to nsv4649522, which has been identified in the 1000 Genomes Project previously.

CNV 15 (chr19:54,555,501–54,560,501) was called a 5 kbp deletion by CNVnator in 4 of the selected individuals (1 homozygous, 3 heterozygous) in the Illumina data. In the SMRT data, it was detected with high sensitivity by all callers. SVIM confirmed its presence, including zygosity, in all samples. Sniffles and pbsv called the variant in accordance with the Illumina data in all but 2 samples. We, therefore, considered CNV 15 validated by the SV callers.

**Table 2** Comparison of the detected CNVs by short-read (illumina) and long-read (PacBio SMRT) sequencing.

| | | Deletions | | Duplications | |
|---|---|---|---|---|---|
| CNV # | Coordinates | Illumina (N)[a] | SMRT ($N_C$, $N_D$, $N_{NA}$)[b] | Illumina (N)[c] | SMRT ($N_C$, $N_D$, $N_{NA}$)[b] |
| 1 | 1:159,018,151–159,018,951 | 0 | 0, 0, 0 | 0 | 0, 0, 0 |
| 2 | 1:179,455,601–179,455,801 | 0 | 0, 0, 0 | 0 | 0, 0, 0 |
| 3 | 3:98,410,601–98,414,801 | 8 | 6, 0, 2 | 0 | 0, 0, 0 |
| 4 | 5:70,391,301–70,394,701 | 1 | 0, 0, 1 | 2 | 0, 2, 0 |
| 5 | 6:31,193,201–31,194,401 | 0 | 0, 0, 0 | 0 | 0, 0, 0 |
| 6 | 6:31,337,891–31,341,891 | 1 | 1, 0, 0 | 0 | 0, 0, 0 |
| 7 | 6:32,450,801–32,453,801 | 0 | 0, 0, 0 | 6 | 4, 2, 0 |
| 8 | 6:32,496,001–32,497,401 | 5 | 0, 1, 4 | 0 | 0, 0, 0 |
| 9 | 6:32,522,201–32,523,601 | 6 | 2, 3, 1 | 0 | 0, 0, 0 |
| 10 | 11:67,330,156–67,332,356 | 1 | 1, 0, 0 | 0 | 0, 0, 0 |
| 11 | 16:28,611,246–28,624,046 | 2 | 1, 0, 1 | 7 | 0, 5, 2[d] |
| 12 | 17:36,387,671–36,399,671 | 0 | 0, 0, 0 | 11 | 3, 8, 0 |
| 13 | 17:39,203,601–39,211,001 | 0 | 0, 0, 0 | 0 | 0, 0, 0 |
| 14 | 19:41,381,791–41,387,591 | 0 | 0, 0, 0 | 0 | 0, 0, 0 |
| 15 | 19:54,555,501–54,560,501 | 4 | 4, 0, 0 | 0 | 0, 0, 0 |

[a] The number of individuals who were heterozygous or homozygous for a deletion (CN < 2) in to the original Illumina data. Not all CNVs were present in individuals passing QC as indicated by N = 0 for both deletions and duplications.
[b] Number of concordant ($N_C$) and discordant ($N_D$) CNV calls, and number of individuals with missing data ($N_{NA}$). A number of concordant individuals represents the number of individuals who had the same CN with Illumina and SMRT data, i.e. CN = 2 with both methods, CN < 2 with both methods, or CN >2 with both methods. A number of discordant individuals represent the number of individuals who had different CN with Illumina and SMRT data. $N_{NA}$: Number of individuals for which the PacBio sequencing results were below QC for the region.
[c] The number of individuals who were heterozygous or homozygous for a duplication (CN > 2) into the original Illumina data.
[d] Individuals with duplications of repetitive elements (4 in total) were considered discordant.
Values are the number of individuals among the 12 that passed QC in the PacBio SMRT sequencing.

A visual inspection of the area also confirmed its presence (Fig. 4). This CNV mapped to dbVar nsv4638371 (Lappalainen et al. 2012).

While most CNVs were either deletions or duplications from the CNVnator calls, 2 CNVs (CNVs 4 and 11) displayed both deletions and duplications (Table 2). For example, in the Illumina data, 2 individuals carried a heterozygous 4,200-bp deletion and 7 individuals displayed a duplication (estimated CN between 2.71 and 5.17) at CNV 11 (chr16: 28,611,246–28,624,046). The duplications could not be replicated in the SMRT data (Supplementary Fig. 1a). Nonetheless, SVIM called several small insertions in this region, corresponding to CN gains reported by CNVnator. In 2 individuals, SVIM identified a tandem duplication of ca. 24 kbp covering the CNV in concordance with CNVnator. However, this was marked as low confidence as there was no read spanning the whole duplication. Additionally, the flanking regions of this CNV exhibited low mapping quality.

CNV 12 (chr17: 36,387,671–36,399,671) was called as a duplication by CNVnator in 11 of the selected individuals (estimated CN between 3.55 and 6.40). However, none of the long-read SV callers detected a large duplication that matched the variant as detected in the Illumina data. Instead, there was evidence for a common large inversion in this area, which showed strongly increased coverage at the breakpoints (Supplementary Fig. 2a). Additionally, 3 individuals had a duplication partly overlapping with this inversion, which we considered concordant with the CNVnator duplication, giving rise to a complex structural rearrangement (Supplementary Fig. 2b).

CNVs 7–9 could not be validated using the stringent criterion requiring 50% reciprocal overlap. However, examining all variant calls overlapping with these CNVs revealed that SVIM identified several large tandem duplications covering the CNVs. In some individuals, there were multiple overlapping duplication calls. Surprisingly, CNVs 8 and 9 were called as strictly deletions by CNVnator. Several individuals had a tandem duplication in this region but no duplication call by CNVnator. However, there were no reads covering the whole variant which indicates it might be a rearrangement instead.

## Discussion

We have identified CNVs in a population-based cohort and tested for association between the CN at CNVs and plasma proteins. We identified a total of 15 CNVs to be associated with 16 proteins, totaling 19 associations. This is clearly a smaller number compared to our previous GWAS to detect associations between SNPs and plasma proteins in the same cohort (Enroth et al. 2018; Höglund et al. 2019). However, this agrees with polymorphic CNVs being numerically fewer than SNPs that are commonly used in GWAS.

Compared to previous studies on CNVs in short-read data, we found fewer variants per sample, especially deletions (Chaisson et al. 2019; Abel et al. 2020). This might be due to our relatively stringent QC requiring high confidence in detected variants. Abel *et al.* specifically highlighted that rare deletions that are shorter than 1 kbp are abundant and prone to false-positive calls. These were therefore more likely to be filtered out during QC because the CNVnator's P values depend on the length of the CNV call. Furthermore, CNVnator appears to misinterpret multiple smaller insertions as larger duplication calls (as seen in e.g. CNV 11), which might have led to a relative overestimation of the number of duplications.

Few previous studies have focused on CNV calling using high-throughput sequencing combined with CNV-phenotype association analysis. A recent study developed a novel method for CNV discovery at population level with 1,457 individuals and tested for association with 275 protein biomarkers (Png et al. 2020). They report 4 significant associations ($P < 1.79 \times 10^{-6}$, resolution: 15 kb). However, none of their significant signals overlapped with our CNVs or were linked to the proteins for which we found associations. By identifying 15 CNVs, our study thereby contributes to increasing the number of CNV-protein associations compared to previous studies.

Overall, we found an unexpectedly large proportion of associations to proteins whose coding genes were distal or on different chromosomes from the CNVs. There could be multiple explanations for this observation. For one, CNVs' effects on gene

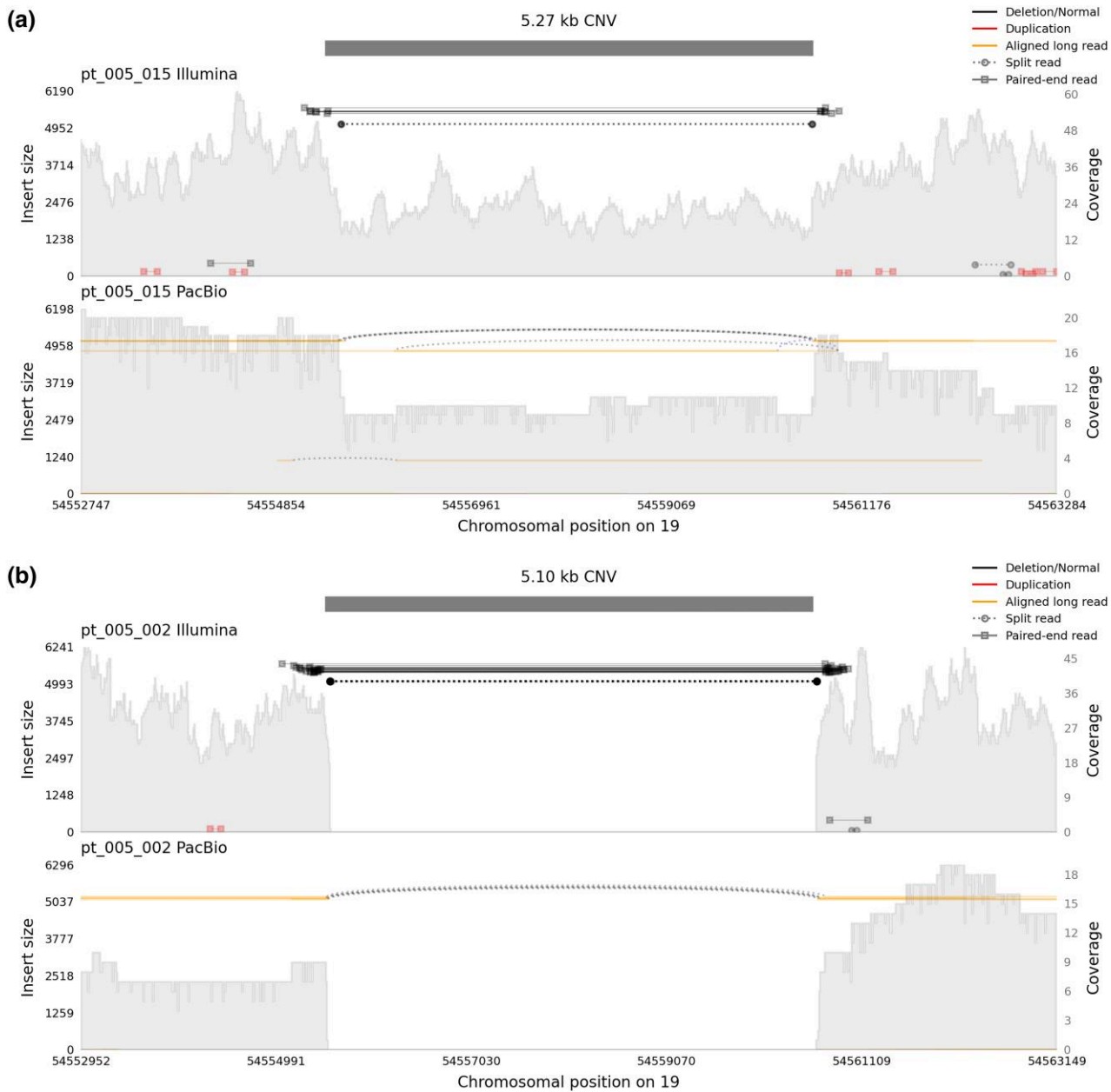

**Fig. 4.** Coverage plot of CNV 15 in 2 individuals. The top shows Illumina data and the bottom SMRT data. a) The same CNV was called a heterozygous deletion (CN 1) in another individual. b) This was called as a homozygous deletion (CN 0). There is very clear evidence of this in both Illumina and SMRT data.

expression are not limited to genes in close proximity. A study using a mouse model found that deletions and duplication of the B2 band of mouse chromosome 11 affected the expression of genes across the whole chromosome (Ricard et al. 2010). Another study investigating the effect of one CNV on chromosome 7 on Williams–Beuren Syndrome found that its presence was associated with differential expression of genes up to 16.7 Mbp away through changed chromatin conformation (Gheldof et al. 2013). Furthermore, *trans*-associations, especially interchromosomal, might be explained by the fact that our study was limited to proteins in blood plasma. Therefore, we were unable to detect differential expression of genes *in cis* whose products do not leave the cell but picked up on changes in protein measurements caused by downstream regulation of pathways (see discussion on CNV 3 below). Finally, some associations could potentially appear to be *in*

*trans* because of misassemblies in the reference genome and might have been considered *in cis* using a more accurate reference assembly, such as T2T-CHM13 (Nurk et al. 2022). However, due to the limited number of associations identified, this difference should be interpreted with care.

CNV 3 (chr3: 98,410,601–98,414,801) was associated with the largest number of proteins, as well as displayed the strongest (lowest *P*-value) association of all identified CNVs. It mapped perfectly to the common structural variant nssv16193243 in dbVar, which has no clinical annotations. While our call agreed with prior literature, the mechanism through which CNV 3 affects the proteins ICAM-2, PD-L2, Siglec-9, CD200R1, and VEGFR-3, is not immediately apparent. The region contains multiple transcription factor binding sites (ENCODE Project Consortium 2012; Davis et al. 2018). The closest downstream gene is *ST3GAL6*, which encodes Type 2

lactosamine alpha-2,3-sialyltransferase (ST3GAL6). This enzyme is involved in the synthesis of Sialyl-Lewis$^X$ (sLeX) and its expression is correlated with levels of sLeX (Kitagawa and Paulson 1993; Carvalho et al. 2010). sLeX is a tetrasaccharide that mediates cell adhesion (Polley et al. 1991). In particular, sulfated sLeX is involved in leukocyte homing and non-modified sLeX plays a role in the recruitment of leukocytes in an inflammatory response and human sperm-egg binding (Munro et al. 1992; Stein et al. 1999; Pang et al. 2011). These functions are closely related to the 5 associated proteins. Intercellular adhesion molecule 2 (ICAM-2) plays an important role in spermatogenesis (Xiao et al. 2013). It is a ligand to Lymphocyte function-associated antigen 1 (LFA-1), which plays a key role in leukocyte adhesion and inflammatory immune response such as cytolysis (Sanchez-Madrid et al. 1982). In fact, ICAM-2 has been found to be upregulated on inflamed pulmonary epithelial cells (Chong et al. 2021). Programmed cell death 1 ligand 2 (PD-L2), on the other hand, is involved in regulating immune response by inhibiting T-cell activation (Latchman et al. 2001). Sialic acid-binding Ig-like lectin 9 (Siglec-9) is a protein of the Siglec family, which mediates cell adhesion by binding to sialic acid (Foussias et al. 2000). Siglec-9 is specific to α-2,3 and α-4,6 linked sialic acid, which means it is able to recognize products of the α-2,3-sialyltransferase ST3GAL6 (Angata and Varki 2000). This relationship hints at the regulation of this protein through availability of ligands produced by ST3GAL6 or that our assay's sensitivity is different for bound and unbound Siglec-9. Cell surface glycoprotein CD200 receptor 1 (CD200R1) is a receptor for CD200, and plays a major role in the inhibition of proinflammatory molecules and cell-endothelium interactions, including adhesion (Wright et al. 2003; Ko et al. 2009). The fact that the aforementioned proteins are involved in processes like leukocyte adhesion and cell recognition or recognize sialic acid, heavily suggests that differential expression of ST3GAL6, caused by the deletion of upstream transcription factor binding sites, affects the prevalence of sLeX. This, in turn, either regulates the proteins' production or affects our assay's sensitivity through their interaction with sLeX. This assessment is supported by the fact that one SNP, rs10935473, which is in strong LD with CNV 3 and is associated with the same proteins (Supplementary Table 5a), is an eQTL for ST3GAL6 and ST3GAL6 Antisense RNA 1 (ST3GAL6-AS1) according to GTeX data (Lonsdale et al. 2013).

CNVs 5–9 are located within the MHC region on chromosome 6. This is a highly variable region, which made the interpretation of our results difficult. CNVs 5–8 were associated with levels of MHC class I polypeptide–related sequence A and B (MIC-AB), which is encoded by the MICA and MICB genes situated in the MHC region. This at least suggests a regulatory effect on these genes (Bahram et al. 1994; Wongfieng et al. 2017). Despite their closeness and association to measurements of the same protein, they were not in LD with each other ($R^2 \leq 0.09$). CNV 9 was associated with Chemokine (C-C motif) ligand 19 (CCL19) measurements, which plays a role in immune response (Kim et al. 1998). Furthermore, the HLA locus is known as a so-called hot spot where CNVs tend to cluster. These can be located in loci relevant to human phenotypes, such as HLA and defensin gene families (Lin and Gokcumen 2019). We could observe these hotspots in our data, as well, where we found numerous duplications and deletions overlapping with multiple CNVs in this locus, including a possible large tandem duplication that could not be unambiguously resolved.

CNV 11 (chr16: 28,611,246–28,624,046) presented as a large duplication and showed an association with ST1A1. While the duplication itself could not be validated using SMRT sequencing, we identified clusters of small insertions in multiple individuals in this region as well as observed low mapping quality in the Illumina data. Both observations might be caused by the repetitive elements in this region, which map well to them (Supplementary Fig. 1b). This provides a possible explanation for the reported CNV. CNVnator might have called these small insertions, which were then merged into a single larger window by the analysis pipeline.

CNV 15 (chr19:54,555,501–54,560,501) was associated with osteoclast-associated immunoglobulin-like receptor (OSCAR), which affects bone metabolism through regulation of osteoclastogenesis (Barrow et al. 2011). The CNV was situated approximately 40 kbp upstream of OSCAR within VSTM1 (V-set and transmembrane domain-containing protein 1), completely covering exons 3, 4, and 5. While we did not find any prior associations in the region covered by the CNV, it is flanked by 2 SNPs (rs10415777, rs4442925) that have been associated with expression of OSCAR (Emilsson et al. 2018; Gudjonsson et al. 2022). These SNPs were in low LD with our deletion (rs10415777: $R^2 = 0.193$, rs4442925: $R^2 = 0.196$), so we considered them independent signals. In addition, CNV 15 is contained in the same topologically associated domain (TAD) as the transcription start site of OSCAR (Rao et al. 2014). These prior findings suggest that CNV 15 affects OSCAR measurements probably through disruption of cis-regulatory elements of the OSCAR locus.

Our long-read sequencing approach showed that CNV calling results from short- and long-read technologies may not be directly comparable in many cases. While the deletions among our target CNVs mostly could be validated, the characterization of the duplications was not immediately clear. For instance, the target CNV on chromosome 16 manifested as many smaller insertions. The binning approach employed by CNVnator might have been responsible for them appearing as a single variant in the Illumina data. SMRT sequencing provided a way to accurately resolve this region.

This study is limited by the exclusion of other SVs such as inversions and translocations. A recent study of haplotype-resolved SVs discovery in the human genome integrated long-read, short-read, strand-specific sequencing technologies and numerous variations calling algorithms in 3 parent-child trios and detected an average of 156 inversions out of 27,622 SVs per sample (Chaisson et al. 2019). The lower frequency of other SVs and lack of long-read strand-specific information cause difficulty in detecting them by only WGS data. Another limitation is that we employed a very stringent significance threshold of $P < 8.22 \times 10^{-10}$, considering each test we performed independent, even when close CNVs were correlated. This reduced our overall sensitivity for associations. Furthermore, there were no new samples taken for long-read sequencing, instead, we used existing biobanked NSPHS samples. This led to degraded DNA quality, as seen in 3 samples which we had to exclude because of low coverage.

In this project, we focus on CNV discovery based on the alignments of short-read WGS reads to the human genome reference. Although the current human genome references (GRCh38 and GRCh37) claim to resolve 99% of the human euchromatic genome, a study constructed a de novo assembly of 2 Swedish genomes by long-read sequences and reported around 10 Mbp novel sequences missing from the GRCh38 mainly located in the centromeric or telomeric regions (Ameur et al. 2018). The misalignments of the reads from the unresolved regions on the current human genome reference can limit the discovery of true signals and lead to false-positive discoveries.

In conclusion, CNVs are associated with measurements of clinically relevant plasma proteins. While we were able to detect CNVs using short reads and corroborate these findings through

association analyses, we found that CNVnator may misrepresent complex large or clusters of smaller variants as simple deletions and duplications. On the other hand, through the use of long-read sequencing, we could accurately resolve these mischaracterized regions, which emphasizes the importance of novel sequencing techniques for future research in this area.

## Data availability

The NSPHS study was approved by the local ethics committee at the University of Uppsala (Regionala Etikprövningsnämnden, Uppsala, Dnr. 2005:325 with approval of extended project period on 2016-03-09). Individual-level genotypes and phenotypes from NSPHS are not publicly available because of their sensitive nature and the scope of the informed consent given by the participants. Data can be made available upon reasonable request as outlined in the supplementary file "Data Access Procedure." The CNV -protein association results generated in this study have been submitted to Zenodo under the record number 7081901 (https://zenodo.org/record/7081901). Further material that supports the findings of this study has been submitted as supplementary tables. The code used to generate the study's results has been submitted to GitHub: https://github.com/AJResearchGroup/cnv-nsphs.

Supplemental material available at GENETICS online.

## Acknowledgments

Sequencing was performed by NGI (National Genomics Infrastructure), Sweden. Protein measurements were carried out by Olink Proteomics AB in Uppsala, Sweden. The computations and data handling were enabled by resources in project sens2016007 provided by the National Academic Infrastructure for Supercomputing in Sweden (NAISS) and the Swedish National Infrastructure for Computing (SNIC) at Uppsala Multidisciplinary Center for Advanced Computational Science (UPPMAX) partially funded by the Swedish Research Council through grant agreements no. 2022-06725 and no. 2018-05973.

## Funding

The NSPHS was funded by the Swedish Foundation for Strategic Research and the Sixth Framework Programme (FP6) of the European Commission. Short read sequencing was funded by the Science for Life Laboratory (SciLifeLab) through the Swedish Genomes Program, which has been made available with support from the Knut and Alice Wallenberg Foundation. Long-read sequencing was supported by Marcus Borgström's and Hedström's foundations. This work was also funded by the Swedish Research Council (2019-01497), the Swedish Heart Lung Foundation (nr. 20200687), and the Swedish Cancer Society (22 2222 Pj).

## Conflict of interest

M.R.A. has performed remunerated consulting services for Olink Proteomics, Uppsala, Sweden. D.S., Z.L., V.L.F., A.A., N.R., and Å.J. declare no competing interests.

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

*Editor: P. Scheet*