## [Peer Review File · Genetics]

Copy number variations and their effect on the plasma proteome

Daniel Schmitz, Zhiwei Li, Valeria Lo Faro, Mathias Rask-Andersen, Adam Ameer, Nima Rafati, and Åsa Johansson

NOTE: The reviews and decision letters are unedited and appear as submitted by the reviewers.

In extremely rare instances and as determined by a Senior Editor or the EIC, portions of a review may be redacted. If a review is signed, the reviewer has agreed to no longer remain anonymous.

The review history appears in chronological order.

Review Timeline:

Submission Date:	2023-04-28
Editorial Decision:	2023-08-07
Resubmission Received:	2023-08-25
Accepted:	2023-09-15

August 2, 2023

GENETICS-2023-306118

Copy number variations and their effect on the plasma proteome

Dear Dr. Schmitz:

Three experts in the field have reviewed your manuscript, and I have read it as well. I am pleased to inform you that, with numerous minor revisions, it is potentially suitable for publication in GENETICS. The reviewers have comments and concerns that need to be addressed in a revised manuscript. You can read their reviews at the end of this email.

It is most important that you address the following in your resubmission. There are numerous clarifications requested by the reviewers, and I agree with their astute observations. Please consider the numerous minor suggestions, which are in some cases critical points for the reader. Also, the following should be considered: number of PacBio participants sequences, address/assess the proportion of trans associations, ensure the supplementary materials are available, and revise the Introduction per Rev. 3's suggestions, particularly adhering to a more concise and linear presentation.

We look forward to receiving your revised manuscript. Please let the editorial office know approximately how long you expect to need for revisions.

Upon resubmission, please include:

1. A clean version of your manuscript;
2. A marked version of your manuscript in which you highlight significant revisions carried out in response to the major points raised by the editor/reviewers (track changes is acceptable if preferred);
3. A detailed response to the editor's/reviewers' comments and to the concerns listed above. Please reference line numbers in this response to aid the editors.

Additionally, please ensure that your resubmission is formatted for GENETICS.

<https://academic.oup.com/genetics/pages/general-instructions>

Follow this link to submit the revised manuscript: Link Not Available

Sincerely,

Paul Scheet
Associate Editor
GENETICS

Approved by:
Anthony Long
Senior Editor
GENETICS

Reviewer #1 (Comments for the Authors (Required)):

In the manuscript "Copy number variations and their effect on the plasma proteome", Schmitz et al. characterize copy number variations in a Swedish cohort of 872 individuals and investigate their association with 438 plasma proteins. Next to the CNV discovery based on short-read WGS, the authors investigate concordance of CNV calling, for the significant CNVs, between short-read WGS with Illumina and long-read SMRT sequencing (PacBio). This study is of general interest to investigators applying short- and long-read sequencing for CNV detection as it demonstrates the technical issues/difficulties with CNV characterization with short-read data using CNVnator and emphasizes the importance of using long-read sequencing for CNV research.

The manuscript is well written and describes the methodologies in detail, though I would like to point out that across the manuscript there is a mismatch in the specified numbers between paragraphs and between paragraphs and numbers (if I understood everything correctly). Examples are listed below:

- Abstract: line 9

Of these, six CNVs could be perfectly validated using long-read sequencing

-> 5 CNVs (CNV 2,3,6,10,15) as stated in Results page 14, line 1 & table 2

- Results page 8, line 3: 15 independent CNVs consisted of 324 of the 200-bp windows .. (Fig 3)

-> In Figure 3: A total of 382 significant 200-bp windows is mentioned

In addition, figure 3 mentions 17 proteins instead of 16

- Results page 9, line 13: $R^2 = 0.339$

-> This is 0.336 in Supplementary Table S4

- Results page 14 line 1: ... of the 14 CNVs that were represented in the resequenced individuals

-> 11 CNVs

- Results page 14 line 4: of ten selected individuals (two heterozygous and eight homozygous)

-> 12 individuals instead of 10? (Table 2)

- Results page 14 line 5: SVIM's calls agreed with CNVator in eight individuals

-> Hence, not all calls are concordant, no validation of this CNV?

- Results page 14 line 12-14: consistently detected by all callers. ... Sniffles and pbsv called the variant in accordance with the Illumina data in all but two samples

-> Hence, not consistently detected by all callers?

- Discussion page 17 line 20: CNV 2 affects the proteins

-> CNV 3?

Other minor remarks:

- Apart from validation of the short-read sequencing results, did long-read sequencing reveal novel CNVs that were not called with short-read seq data?

- Abstract: Line 6: (N=1,021) using short- and long-read whole genome sequencing. Long-read WGS was only applied on 15 individuals, this sentence gives the impression that long-read sequencing was applied on the complete cohort.

- Introduction:

Figure 1: suggestion to also add '12 individuals' before the validation step

Page 6 line 5: res-sequencing -> resequencing

- Results

Page 8 line 14: (Supplementary Table S1) of CNV1

-> Supplementary Table S4 instead of S1?

-> in Supplementary Table S4, Supplementary Table S2A, B,.. are mentioned instead of S4

Page 13 line 15: Among the 15 CNVs, 11 were called with a CN different from two in at least one of the twelve individuals we selected for resequencing.

Hence, only 11 CNVs could be investigated for replication with long-read as the twelve individuals did not carry the other 4 CNVs.

Were the 3 individuals that failed SMRT sequencing QC carrier of these 4 CNVs?

For the selection of individuals for resequencing with SMRT, individuals were selected who displayed CNs different from 2 for the largest number of the protein associated CNVs; but ideally you would like to be sure that the 'replication' cohort comprises all 15 CNVs that were identified with short-read seq?

Supplementary figures (S1A, S2A, S2B): would it be possible that these are missing?

Supplementary Table 6 & Discussion page 20 line 21-22:

Apart from the 3 individuals with < 50% of the reads from high quality, there were also 3 individuals with 'borderline' % of reads from high quality (close to 50%).

Reviewer #2 (Comments for the Authors (Required)):

The manuscript seeks to explore potential relationships between germline copy-number variants and levels of plasma proteins. Here they identify CNVs from a set of ~1,000 individuals using short-read whole-genome sequencing and assess correlations between CNV genotype and plasma proteins. This manuscript extends on previous work and data from this group assessing correlations between plasma protein levels and genotype detected by SNP array (Enroth et al, 2014; Enroth et al 2018) or WGS (Kierczak et al 2022), here using previously published WGS data from the same sample set (Hoglund et al, 2019) to identify novel correlations between larger CNVs and plasma protein level. In total they identify 19 proteins at 15 loci with significant associations with CNV genotype. Of these loci, CNVs were validated in a fair proportion using long-read sequencing.

Overall this study is a logical extension of the authors past work and one of only a few studies examining associations between CNVs and plasma proteins and suitable for the journal and meets the criteria for publication. Overall the manuscript is well written and doesn't seem to have any fundamental issues. There are a number of minor points that the authors could address that could help with clarity, accuracy, or discussion:

The authors note in the methods that they correct for the total number of tests (60,814,115 tests $\rightarrow P < 8.22 \times 10^{-10}$). In the manuscript they note they are testing 184,182 CNV bins versus 438 plasma proteins. Shouldn't this be 80,671,716 tests (184182 x 438) and $P < 6.20 \times 10^{-10}$?

It would be helpful to clarify in the methods what the final CN value is actually used in association testing. As best I can tell without having specifically used CNVnoator, CNVnator output gives a CNV call along with a normalized read-depth (RD) value. I'm assuming that this estimated RD value is what is being used for any samples with a CNV called at a specific bin that passes the q0 filter (e.g. CN isn't a discrete copy number - 0/1/2/3/etc, unless no CNV is called in which case CN=2)? The CNVnator data in Table S7 suggests this is likely the case; Fig 2 also shows estimated copy-numbers. It would be helpful to a reader to clarify.

Should the reference to Table S1 on line 14 of page 8 refer to Table S4 instead? That's the table that seems to show independence of same-chromosome loci?

Page 17 line 19-20 refers to "CNV 2", should this be "CNV 3" instead?

The manuscript and Table S6 seem to be inconsistent about the number of samples sequenced via PacBio. The manuscript notes 15 samples sequenced via PacBio, with three failing QC (<50% of reads of high quality). Table S6 seems to show 17 samples, with 3 samples with <50% high quality reads. So was the total number of individuals used 14 rather than 12?

In the discussion of the CNV3 locus (p17-18), the authors posit a reasonable hypothesis. Two thoughts that might strengthen this discussion. (1) what initially jumped out on Table 1 was that of the 5 associations at this locus, one was "cis" (CD200R1) while the remaining were trans, which often suggests the trans association may be explained by or mediated by the cis association, and CD200R1 is known to inhibit the expression of proinflammatory molecules. Upon closer inspection this seems somewhat implausible - the distance from the locus to the gene is 4-5 Mb which is perhaps too far and likely not within the same TAD - e.g. these aren't necessarily cis- in gene regulatory terms? (I guess one could test for mediation?). (2) the authors identified a SNP in strong LD from the WGS-based GWAS (rs10935473, $r^2 = 0.78$) - this SNP seems to be a rather significant multi-tissue eQTL for ST3GAL6 (and ST3GAL6-AS1) in GTEx, which I think provides a bit further support for the authors hypothesis that these associations may be mediated by ST3GAL6 biology.

I'm wondering if the authors might comment on the proportion of associations that are trans in these data, and perhaps contrast with what's seen with the SNV-based GWAS data? A quick inspection of Table 1 suggests most of these CNV loci are cis-associations with the protein target, but in fact a number of these may be on the same chromosome but have the CNV and target separated by a larger distance (>1 Mb) typically considered cis in gene regulatory terms. 11/19 CNV-protein associations are trans or separated by more than 1Mb; such associations are found at 7/15 loci. This proportion of significant trans associations seems larger than one might expect, and larger than what is observed in the GWAS from Kierczak et al 2022. Any thoughts as to why these trans associations with CNVs seem so common?

I didn't seem to be able to access the Supplementary Figures on the manuscript submission system?

Reviewer #3 (Comments for the Authors (Required)):

The authors conducted association analyses of CNVs and plasma proteins findings several associations, many of which are cis associations. They have done an excellent job detailing their approach in the methods section and conducting validation of their initial findings. There are a few areas that the authors should address to better present their work.

- 1- The authors should streamline the introduction and focus on presenting their ideas more concisely and in a linear manner.
- 2- The authors should change line 14 (page 3) and instead of stating that CNVs "cause dosage imbalance of the genes involved" they should make it clear that CNVs "often cause dosage imbalance of the genes involved" This point needs a reference.
- 3- On page 6 line 16 and top of page 7, the authors report the average number of deletions and duplications found per individual. How do these compare to previous studies? If these are different, why would that be?
- 4- Page 7 line 5, it is not clear what the authors mean by a "less stringent filter". A specific value would be helpful.
- 5- Page 23 (methods) line 22. Is the CN for a window treated as a discrete or continuous value? It would help the reader if both the methods and results clearly explained how CNVs/CNs were modeled for the association analyses.
- 6- Page 24 line 4. The multiple testing adjustment is very stringent/conservative, the authors may want to consider mentioning this point.
- 7- Page 25 line 13, the requirement of 50% reciprocal overlap is stringent. Do the results change if the threshold is lowered?

Associate Editor Comments:

Point-to-Point Response to Reviewers' Comments

Reviewer #1

In the manuscript "Copy number variations and their effect on the plasma proteome", Schmitz et al. characterize copy number variations in a Swedish cohort of 872 individuals and investigate their association with 438 plasma proteins. Next to the CNV discovery based on short-read WGS, the authors investigate concordance of CNV calling, for the significant CNVs, between short-read WGS with Illumina and long-read SMRT sequencing (PacBio). This study is of general interest to investigators applying short- and long-read sequencing for CNV detection as it demonstrates the technical issues/difficulties with CNV characterization with short-read data using CNVnator and emphasizes the importance of using long-read sequencing for CNV research.

The manuscript is well written and describes the methodologies in detail, though I would like to point out that across the manuscript there is a mismatch in the specified numbers between paragraphs and between paragraphs and numbers (if I understood everything correctly). Examples are listed below:

Response: We thank the reviewer for these important comments. We have addressed all the points as described in detail below.

Abstract: line 9: Of these, six CNVs could be perfectly validated using long-read sequencing
-> 5 CNVs (CNV 2,3,6,10,15) as stated in Results page 14, line 1 & table 2

Response: We apologize for this mistake, the text has now been corrected (page: 2, line: 9-10)

Results page 8, line 3: 15 independent CNVs consisted of 324 of the 200-bp windows (Fig 3)
-> In Figure 3: A total of 382 significant 200-bp windows is mentioned

In addition, figure 3 mentions 17 proteins instead of 16

Response: We apologize for this mistake, the text has now been corrected (p. 10 l. 15-16, p. 32 l. 19-21)

Results page 9, line 13: $R^2 = 0.339$

-> This is 0.336 in Supplementary Table S4

Response: We apologize for this mistake; the value has now been corrected (Supplementary Table S4c)

Results page 14 line 1: ... of the 14 CNVs that were represented in the resequenced individuals -> 11 CNVs

Response: We apologize for this mistake; the text has now been corrected (p.13 l.3-4).

Results page 14 line 4: of ten selected individuals (two heterozygous and eight homozygous)
-> 12 individuals instead of 10? (Table 2)

Response: We apologize for this mistake; the text has now been corrected (p.13 l.8-9).

Results page 14 line 5: SVIM's calls agreed with CNVator in eight individuals

-> Hence, not all calls are concordant, no validation of this CNV?

Response: We apologize for not including a clear definition of what we considered as a validation. We have now clarified this in the text to make it clear why we considered these variants validated by SVIM (p.13 l.10-12)

Results page 14 line 12-14: consistently detected by all callers. ... Sniffles and pbsv called the variant in accordance with the Illumina data in all but two samples

-> Hence, not consistently detected by all callers?

Response: We added a clarification to the manuscript including our verdict that the variants were considered validated (p.13 l.21-22)

Discussion page 17 line 20: CNV 2 affects the proteins

-> CNV 3?

Response: We apologize for this mistake; the text has now been corrected (p. 17 l. 1).

Apart from validation of the short-read sequencing results, did long-read sequencing reveal novel CNVs that were not called with short-read seq data?

Response: As the long-read sequencing was intended to act as a validation study and not for discovery, we did not ensure sufficient quality of novel variant calls from this approach. Additionally, only pbsv reported CNV calls as such and allowed for joint genotyping. We performed this analysis and found 616 CNVs passing filters in total.

Abstract: Line 6: (N=1,021) using short- and long-read whole genome sequencing. Long-read WGS was only applied on 15 individuals, this sentence gives the impression that long-read sequencing was applied on the complete cohort.

Response: We have corrected this ambiguity in the manuscript (p.2 l.5-8).

Introduction:

Figure 1: suggestion to also add '12 individuals' before the validation step

Page 6 line 5: res-sequencing -> resequencing

Response: We thank the reviewer for pointing out these mistakes and have corrected them in the manuscript (p. 32 l.5).

Results

Page 8 line 14: (Supplementary Table S1) of CNV1

-> Supplementary Table S4 instead of S1?

-> in Supplementary Table S4, Supplementary Table S2A, B,.. are mentioned instead of S4

Response: We have corrected these mistakes (p. 11 l.7, Supplementary Table S4).

Page 13 line 15: Among the 15 CNVs, 11 were called with a CN different from two in at least one of the twelve individuals we selected for resequencing.

Hence, only 11 CNVs could be investigated for replication with long-read as the twelve individuals did not carry the other 4 CNVs.

Response: We have changed these numbers to agree with the remainder of the manuscript. (p.13 l.3-4).

Were the 3 individuals that failed SMRT sequencing QC carrier of these 4 CNVs? For the selection of individuals for resequencing with SMRT, individuals were selected who displayed CNs different from 2 for the largest number of the protein associated CNVs; but ideally you would like to be sure that the 'replication' cohort comprises all 15 CNVs that were identified with short-read seq?

Response: Unfortunately, for some CNVs all individuals that passed QC in the long-read sequencing were carriers of the reference allele (CN=2). However, we also disregarded some regions e.g., the HLA region when selecting individuals for resequencing, due to its complexity and therefore there were some additional regions where no individuals had the alternative call. This has now been clarified in the text (p.9 l.8-11) and in a footnote to Table 2 (p.35 l.4-5).

Supplementary figures (S1A, S2A, S2B): would it be possible that these are missing?

Response: We apologize for this mistake. We have included the supplementary figures in the resubmission.

Supplementary Table 6 & Discussion page 20 line 21-22:

Apart from the 3 individuals with < 50% of the reads from high quality, there were also 3 individuals with 'borderline' % of reads from high quality (close to 50%).

Response: We thank the reviewer for pointing this out as this revealed this section to be unclear. The remark about the fraction of high-quality reads was mainly exemplary and we failed to add that we excluded individuals with a mean coverage less than 10x (p.9 l.18-19, p.12 l.20-22, p. 20 l.13-14).

Reviewer #2

The manuscript seeks to explore potential relationships between germline copy-number variants and levels of plasma proteins. Here they identify CNVs from a set of ~1,000 individuals using short-read whole-genome sequencing and assess correlations between CNV genotype and plasma proteins. This manuscript extends on previous work and data from this group assessing correlations between plasma protein levels and genotype detected by SNP array (Enroth et al, 2014; Enroth et al 2018) or WGS (Kierczak et al 2022), here using previously published WGS data from the same sample set (Hoglund et al, 2019) to identify novel correlations between larger CNVs and plasma protein level. In total they identify 19 proteins at 15 loci with significant associations with CNV genotype. Of these loci, CNVs were validated in a fair proportion using long-read sequencing.

Overall this study is a logical extension of the authors past work and one of only a few studies examining associations between CNVs and plasma proteins and suitable for the journal and meets the criteria for publication. Overall the manuscript is well written and doesn't seem to have any fundamental issues. There are a number of minor points that the authors could address that could help with clarity, accuracy, or discussion:

Response: We thank the reviewer for providing these valuable points. We have addressed all the points as described in detail below

The authors note in the methods that they correct for the total number of tests (60,814,115 tests -> $P < 8.22 \times 10^{-10}$). In the manuscript they note they are testing 184,182 CNV bins versus 438 plasma proteins. Shouldn't this be 80,671,716 tests (184182 x 438) and $P < 6.20 \times 10^{-10}$?

Response: The final number of tests for which we adjusted was the number of tests actually performed as we did not run the model for combinations of CNVs and proteins where less than three individuals had CNs different from 2 and protein measurements. We have clarified this in the manuscript (p. 8 l.5-9).

It would be helpful to clarify in the methods what the final CN value is actually used in association testing. As best I can tell without having specifically used CNVnator, CNVnator output gives a CNV call along with a normalized read-depth (RD) value. I'm assuming that this estimated RD value is what is being used for any samples with a CNV called at a specific bin that passes the q0 filter (e.g. CN isn't a discrete copy number - 0/1/2/3/etc, unless no CNV is called in which case CN=2)? The CNVnator data in Table S7 suggests this is likely the case; Fig 2 also shows estimated copy-numbers. It would be helpful to a reader to clarify.

Response: CN estimates reported by CNVnator are continuous values that quantify the factor by which the read depth within the CNV differs from the depth around it. We have added a clarification in the manuscript (p.6 l.15-18, p.7 l.22-23).

Should the reference to Table S1 on line 14 of page 8 refer to Table S4 instead? That's the table that seems to show independence of same-chromosome loci?

Response: We thank the reviewer for pointing out this inconsistency and have corrected this mistake (p. 11 l. 7).

Page 17 line 19-20 refers to "CNV 2", should this be "CNV 3" instead?

Response: We have now corrected this number (p. 17 l. 1).

The manuscript and Table S6 seem to be inconsistent about the number of samples sequenced via PacBio. The manuscript notes 15 samples sequenced via PacBio, with three failing QC (<50% of reads of high quality). Table S6 seems to show 17 samples, with 3 samples with <50% high quality reads. So was the total number of individuals used 14 rather than 12?

Response: The samples pt_013_001 and pt_013_002 were additional sequencing runs of individuals pt_005_001 and pt_005_002 with a different protocol. We opted not to include this data in our analysis as not to introduce batch effects into our data. We have removed them from Supplementary Table S6.

In the discussion of the CNV3 locus (p17-18), the authors posit a reasonable hypothesis. Two thoughts that might strengthen this discussion. (1) what initially jumped out on Table 1 was that of the 5 associations at this locus, one was "cis" (CD200R1) while the remaining were trans, which often suggests the trans association may be explained by or mediated by the cis association, and CD200R1 is known to inhibit the expression of proinflammatory molecules. Upon closer inspection this seems somewhat implausible - the distance from the locus to the gene is 4-5 Mb which is perhaps too far and likely not within the same TAD - e.g. these aren't necessarily cis- in gene regulatory terms? (I guess one could test for mediation?). (2) the authors identified a SNP in strong LD from the WGS-based GWAS (rs10935473, $r^2 = 0.78$) - this SNP seems to be a rather significant multi-tissue eQTL for

ST3GAL6 (and ST3GAL6-AS1) in GTEX, which I think provides a bit further support for the authors hypothesis that these associations may be mediated by ST3GAL6 biology.

Response: We thank the reviewer for providing these insights. Regarding 1) We ran the model again for this particular CNV, adjusting for measurements of CD200R1. All associations stayed significant and effect direction was preserved (see table below), which suggests that CD200R1 does not mediate the association with the other proteins. We have also revised the annotation of “Cis” regions in Table 1 to only be CNV regions within 2 Mb from the encoding gene, and not all regions on the same chromosome. For that reason, CD200R1 is no longer annotated as a Cis region and the table below is therefore not included in the manuscript. Regarding 2) We have added a remark about the SNP association to the discussion in the manuscript (p.18 l.6-8)

Protein	Beta	SE	P value
PD-L2	0.38345309	0.04444305	3.42E-17
ICAM-2	0.64843304	0.04067079	5.82E-50
VEGFR-3	0.5809207	0.04301156	1.82E-37
Siglec-9	0.67141426	0.04268022	1.61E-48

I'm wondering if the authors might comment on the proportion of associations that are trans in these data, and perhaps contrast with what's seen with the SNV-based GWAS data? A quick inspection of Table 1 suggests most of these CNV loci are cis-associations with the protein target, but in fact a number of these may be on the same chromosome but have the CNV and target separated by a larger distance (>1 Mb) typically considered cis in gene regulatory terms. 11/19 CNV-protein associations are trans or separated by more than 1Mb; such associations are found at 7/15 loci. This proportion of significant trans associations seems larger than one might expect, and larger than what is observed in the GWAS from Kierczak et al 2022. Any thoughts as to why these trans associations with CNVs seem so common?

Response: We have revised the *cis/trans* classification to be in line with Kierczak et al 2022 i.e., a distance < 2 Mbp was considered *in cis*. The abundance of *trans*-associations might have many reasons. For one, CNVs have been shown to be able to affect differential expression over long-ranges, partly through induced changes in chromatin conformation. Additionally, the detected CNVs might actually regulate the expression of genes *in cis*. However, we were unable to detect these changes directly as we were only able to measure proteins released into the bloodstream. We addressed these points in the results and the discussion sections of the manuscript (p 10 l.-21, p.16 l.3-19).

I didn't seem to be able to access the Supplementary Figures on the manuscript submission system?

Response: We apologize for this mistake. We have included the supplementary figure in the revised manuscript.

Reviewer #3

The authors conducted association analyses of CNVs and plasma proteins findings several associations, many of which are cis associations. They have done an excellent job detailing their approach in the methods section and conducting validation of their initial findings.

There are a few areas that the authors should address to better present their work.

Response: We thank the reviewer for their comments. We have addressed all the points as described in detail below

The authors should streamline the introduction and focus on presenting their ideas more concisely and in a linear manner.

Response: We have revised the introduction according to the input by this reviewer (p. 3 l.3 - p.5 l.11)

The authors should change line 14 (page 3) and instead of stating that CNVs "cause dosage imbalance of the genes involved" they should make it clear that CNVs "often cause dosage imbalance of the genes involved" This point needs a reference.

Response: We thank the reviewer for the suggestion, which we have implemented in the manuscript (p. 3 l. 14-16).

On page 6 line 16 and top of page 7, the authors report the average number of deletions and duplications found per individual. How do these compare to previous studies? If these are different, why would that be?

Response: We found fewer CNVs per individual on average than previous studies. This is mainly driven by a reduced sensitivity for deletions, as the number of duplications per individual. This might be driven by our stringent QC, which is more likely to remove short, low-confidence variants, which tend to be skewed towards deletions and CNVnator overestimating the number of duplications by merging small insertions within close proximity. We have added these points to the manuscript (p. 10 l.7-8, p. 15 l. 10-17).

Page 7 line 5, it is not clear what the authors mean by a "less stringent filter". A specific value would be helpful.

Response: We have added a clarification in the manuscript (p.7 l.12-14)

Page 23 (methods) line 22. Is the CN for a window treated as a discrete or continuous value? It would help the reader if both the methods and results clearly explained how CNVs/CNs were modeled for the association analyses.

Response: CNs as calculated by CNVnator are continuous values representing the factor by which the read depth within the CNV differs from its flanking regions. We have added a clarification the manuscript (p.6 l.15-18, p.7 l.22-23).

Page 24 line 4. The multiple testing adjustment is very stringent/conservative, the authors may want to consider mentioning this point.

Response: We agree with the reviewer and have mentioned this as a limitation in the manuscript (p.20 l.9-11).

Page 25 line 13, the requirement of 50% reciprocal overlap is stringent. Do the results change if the threshold is lowered?

Response: Considering all variant calls that imply a CN change in the same direction (without requiring 50% reciprocal overlap) as the CNV call, we found larger duplications and deletions spanning multiple CNVs in some individuals, mainly on chromosome 6. We have added this observation to the manuscript (p.9 l.22, p.14 l.7-9, p.14 l.16-17, p.14 l.19-p.15 l.2, p.18 l.19-21)

September 13, 2023
RE: GENETICS-2023-306425

Mr. Daniel Schmitz
Uppsala Universitet
Department of Immunology, Genetics and Pathology
Box 815
Uppsala, N/A 751 08
Sweden

Dear Dr. Schmitz:

Congratulations! We are delighted to inform you that your manuscript entitled "Copy number variations and their effect on the plasma proteome" is acceptable for publication in GENETICS. Many thanks for submitting your research to the journal.

To Proceed to Production:

1. Format your article according to GENETICS style, as discussed at <https://academic.oup.com/genetics/pages/general-instructions>, and upload your final files at <https://genetics.msubmit.net>.
2. Your manuscript will be published as-is (unedited-as submitted, reviewed, and accepted) at the GENETICS website as an Advanced Access article and deposited into PubMed shortly after receipt of source files and the completed license to publish. Please notify sourcefiles@thegsajournals.org if you do not wish to publish your article via Advanced Access.
3. We invite you to submit an original color figure related to your paper for consideration as cover art. Please email your submission to the editorial office or upload it with your final files. You can submit a small-sized image for evaluation, and if selected, the final image must be a TIFF file 2513px wide by 3263px high (8.375 by 10.875 inches; resolution of 600ppi). Please avoid graphs and small type.

If you have any questions or encounter any problems while uploading your accepted manuscript files, please email the editorial office at sourcefiles@thegsajournals.org.

Sincerely,

Paul Scheet
Associate Editor
GENETICS

Approved by:
Anthony Long
Senior Editor
GENETICS

note: Please add jnls.author.support@oup.com and genetics.oup@kwglobal.com (or the domains @oup.com and @kwglobal.com) to your email program's "safe senders" list. You will be contacted by both at various points during the production process.